# Metered-Dose Inhaler Spacer with Integrated Spirometer for Home-Based Asthma Monitoring and Drug Uptake

**DOI:** 10.3390/bioengineering11060552

**Published:** 2024-05-29

**Authors:** Cheuk-Yan Au, Kelleen J. X. Koh, Hui Fang Lim, Ali Asgar Saleem Bhagat

**Affiliations:** 1Institute for Health Innovation & Technology (iHealthtech), National University of Singapore (NUS) MD6, 14 Medical Drive, #14-01, Singapore 117599, Singapore; cy.au@nus.edu.sg (C.-Y.A.); kelleenkoh@u.nus.edu (K.J.X.K.); 2Division of Respiratory and Critical Care Medicine, Department of Medicine, National University Health System, Singapore 119074, Singapore; limhuifang01@gmail.com; 3Department of Biomedical Engineering, National University of Singapore (NUS), 4 Engineering Drive 3, Singapore 117583, Singapore

**Keywords:** asthma, asthma control, spirometry, inhaler technique, inhaler spacer, medical device

## Abstract

This work introduces Spiromni, a single device incorporating three different pressurised metered-dose inhaler (pMDI) accessories: a pMDI spacer, an electronic monitoring device (EMD), and a spirometer. While there are devices made to individually address the issues of technique, adherence and monitoring, respectively, for asthma patients as laid out in the Global Initiative for Asthma’s (GINA) global strategy for asthma management and prevention, Spiromni was designed to address all three issues using a single, combination device. Spiromni addresses the key challenge of measuring both inhalation and exhalation profiles, which are different by an order of magnitude. Moreover, the innovative design prevents exhalation from entering the spacer chamber and prevents medication loss during inhalation using umbrella valves without a loss in flow velocity. Apart from recording the peak exhalation flow rate, data from the sensors allow us to extract other key lung volume and capacities measures similar to a medical pulmonary function test. We believe this low-cost portable multi-functional device will benefit both asthma patients and clinicians in the management of the disease.

## 1. Introduction

Asthma is a pervasive, incurable chronic lung condition affecting individuals of all ages globally; up to 339 million people. This health issue has substantial repercussions, accounting for 21.6 million Disability-Adjusted Life Years (DALYs) and a mortality rate of 5.8 per 100,000 individuals [1]. Notably, Singapore reports a mortality rate three times higher than the worldwide average [2], underscoring the urgency of addressing and mitigating its impact on public health. Distinct symptoms of asthma include wheezing during breathing, breathlessness, persistent coughing, and tightness in the chest [3], and these symptoms could escalate and become life-threatening [4]. Patients are encouraged to adopt a comprehensive approach outlined by the Global Initiative for Asthma (GINA) [5] of identifying and avoiding potential asthmatic triggers [6], following a strict regimental adherence to prescribed medications, consistently monitoring their respiratory health, and seeking timely medical intervention when necessary to achieve effective asthma control. A key component of the approach involves the use of pressurised metered-dose inhalers (pMDIs), which facilitate the administration of drugs such as glucocorticosteroids at regular intervals. Critical elements of the GINA strategy also incorporates routine follow-up consultations. These consultations help to assess and ensure patient adherence and compliance, conduct essential tests to measure lung functions, as well as review the effectiveness of the prescribed treatment in controlling asthma symptoms [5].

Assessing adherence is highly reliant on patients’ self-reporting, typically through the use of questionnaires for their cost-effectiveness and ease of implementation in clinical settings [7]. However, these methods are highly inaccurate, as patients tend to overstate their level of adherence [8]. In recent years, the rapid progress of technology has led to a proliferation of commercial digital inhalers and pMDI electronic monitoring devices (EMDs) entering the market. Notable examples include the Teva Digihaler^®^ [9] and the Hailie Sensor^®^ [10]. With the use of these devices, impartial real-time data of inhaler usage can be collected, enhancing the precision of the adherence assessment of patients. At the same time, its supportive features—such as audio–visual reminders, guided usage and feedback to patients—can assist the patient in the usage of the pMDI. These advancements represents a pivotal shift, potentially addressing the problems associated with the accuracy and reliability of the traditional self-reporting method when assessing adherence [11]. Similar to adherence, patients who are self-reporting their lung condition between consultations could poorly assess their condition due to a poor recognition and perception of the severity of their symptoms. Hence, there is a need to rely on pulmonary function test parameters that are objective in nature, such as Forced expiratory volume in the first second (FEV1), Tiffeneau–Pinelli index (FEV1/FVC), and Peak Expiratory Flow Rate (PEFR) that are obtained through the forced expiratory manoeuvre. These results significantly enhance clinicians’ confidence in diagnosing and adjusting treatments [5]. Clinicians face challenges in monitoring patients’ lung function and variability over a continuous period of time [12]. Portable spirometers, such as CMI Health Spirolink^®^ [13] and MIR Spirobank Smart [14], offer a solution by enabling home-based monitoring. This allows for more frequent measurements of the critical parameters, providing valuable insights into the lung functions, and facilitating a more comprehensive understanding of patients’ respiratory health. The final and most significant key to an effective treatment is the inhaler technique the patient uses. Poor coordination between firing the pMDI and inhaling the drug could drastically reduce the amount of drugs deposited in patients’ lungs [15] and is a major issue for inhaler users [16]. The use of optional spacer devices attached onto pMDIs overcomes the coordination issue but comes with the disadvantage of being bulky [15]. Portable spirometers, spacer devices and pMDI EMDs are currently all individually available as commercial products ready for purchase; however, owning and carrying around multiple devices would be cumbersome and inconvenient for the patients.

Currently, there is no single device on the market that can perform these three functions. We propose a portable, easy-to-use, multi-functional spacer device that is capable of measuring the inhalation profile of the patient, as well as performing as a portable spirometer. This paper reports the feasibility of measuring both inhalation and exhalation profiles that is needed for both a spirometer and EMD within a single device, despite the difference in magnitude of 10 times between their flow rates.

## 2. Materials and Methods

### 2.1. Spiromni

The design of Spiromni (Figure 1i) consists in 2 intersected cylinders embedded with pneumotachographs (PNTs) on each cylinder measuring airflow via pressure differential across a flow-resistive element or screen. The two main considerations when designing Spiromni were: (a) to accommodate the range of the inhalation and exhalation flow rates for sensor measurements, and (b) to minimise the turbulence and dead volume for accurate flow rate measurements, as well as to optimise drug delivery from the spacer to the user. One of the cylinder is elongated to function as a spacer. The device weighs 80 g and was designed for measuring flow rates between 24 and 120 litres per minute (L/min) for inhalation, and between 48 and 600 L/min for exhalation. The inhalation flow rate values were selected to accommodate measurement from 24 to 90 L/min, the optimal inspiratory flow rate (OIFR) for adequate deposition of drugs in the lungs with a spacer device [17]. The exhalation values were chosen to suit the measurable peak expiratory flow rate (PEFR) at 600 L/min for the Southeast Asian demographic [18,19].

Spiromni consists of three separate 3D-printed parts: an end cap for interfacing with the pMDI, the spacer section, and spirometer section with a mouthpiece for the user (Figure 1ii) designed using SOLIDWORKS 2023. The PNT screens across the spacer and spirometer sections were augmented with umbrella valves over the PNT screen (Figure 1iii) so they could function as a one-way valve to divert exhaled air out of the device (Figure 2i) and draw air from the spacer when inhaling, respectively (Figure 2ii,iii). The PNT screens are 1 mm in thickness and made from a series of concentric annular cutouts to determine the resistance of flow when the umbrella valve is open. Between each PNT screen, a Sensiron SDP31 differential pressure sensor measures the pressure differential. In the event where the user inhales, an SDP31 sensor, Isensor, reads the pressure differential, while another SDP31 sensor, Esensor, reads the pressure difference between the spirometer section and the atmosphere during exhalation. Both Isensor and Esensor were read using a Sparkfun Thing Plus’s ESP32 microcontroller at a rate of 20 Hz with integrated Bluetooth Low Energy (BLE) for transmitting the data to a device.

The spacer section was designed to have a capacity of 100 mL cylindrical section and is 55 mm in length. It is classified as a medium-volume spacer [20]. The end cap has a sheet of silicone cut and adhered with silicone sealant, to create a seal when interfacing with a pMDI actuator mouthpiece (Figure 1ii).

### 2.2. Simulation

The simulations were done in SOLIDWORKS 2023 flow simulation package using the internal pipe computational fluid dynamics (CFD) analysis. All the parts are mated together into a rigid assembly and lids are placed on all openings to create a water-tight model. The mesh was generated automatically with adiabatic walls, 0 mm roughness and the flow were computed with no gravity effects. The mouthpiece lid was set to be a uniform inlet mass flow for exhalation, and as an outlet mass flow to mimic inhalation. The PNT screens were set to be exposed to the atmosphere with respect to the airflow as described in the flow diagram (Figure 2) for exhalation and inhalation, respectively. An additional lid was placed at the inhaler interface to measure velocity for inhalation simulation only.

The median flow rate of 5 litres per second (L/s), or 300 L/min was used to model initially with 1 L/s and 10 L/s (60 L/min and 600 L/min) used to see the limits for exhalation. For inhalation, 1 L/s (60 L/min) was used initially and 0.2 L/s and 2 L/s (12 L/min and 120 L/min) for the limits. All units were in L/s to comply with the CFD input parameters. Models were iteratively altered to have the least amount of turbulence across all the specified ranges to maximise flow across the differential pressure sensors. The main parameters analysed were the average flow velocity and vectors.

### 2.3. Sensor Calibration

The two Sensirion SDP31 differential pressure sensors, Esensor and Isensor, were calibrated to accurately collect the flow rate during inhalation and exhalation due to the complex shape of Spiromni. Instead of using syringes [20] or bags [21] of fixed volumes, a G-TACK 1HP air compressor with an attached 12 L air tank and the Omega FMA-1613A industrial flow rate sensor was used to calibrate the device. The setup had a flow control valve to release air from the compressor tank at a series of different flow rates (Figure 3). The flow rate measured in L/min from the industrial flow rate sensor and the pressure sensors values in pascals (Pa) were collected in sync using BLE onto a computer using pySerial and Bleak packages on Python 3.9.10.

#### 2.3.1. Esensor Calibration

The exhalation calibration setup has the air tank connected to the mouthpiece of Spiromni with a ball valve and a flow control valve. During calibration, the tank was fully compressed to 700 KPa, and the flow control valve set to output the intended flow rate. Esensor and flow rate sensor data were collected as the ball valve was opened fully as quickly as possible and then closed. Each test took around 5 s and it mimicked the forced expiratory manoeuvre and the peak of the each set of data was collected and plotted. At least 3 successful sets of data were taken in steps of 20 L/min until 700 L/min. Beyond 700 L/min, the calibration setup was unable to provide stable flow for at least 2 s.

#### 2.3.2. Isensor Calibration

In the inhalation calibration setup, the air tank was connected to the pMDI spacer end cap to use positive flow rate to mimic inhalation. This is due to the lack of a device that could generate constant and adequate negative pressure and flow rate. The calibration steps were similar to the exhalation calibration, with 3 sets of data taken in steps of 10 L/min until 100 L/min.

## 3. Results

### 3.1. Flow Profile Simulation

During the design process, the flow patterns within Spiromni during exhalation and inhalation were constantly monitored to minimise turbulence as much as possible. The flow simulation revealed that during exhalation, turbulence was found at the intersection of the two cylinders of the spirometer section (Figure 4i), and that there is also a constant 8% loss in flow velocity from the tip of the mouthpiece to Esensor. During inhalation, the flow is laminar, with a 12% loss in the flow velocity from the mouthpiece to Isensor and its PNT screen, and a 43% loss at the end cap (Figure 4ii).

### 3.2. Sensor Calibration Results

With the data collected from the calibration experiment, a calibration line/curve could be drawn. Esensor exhibits a linear relationship, with variance, with increasing flow rate (Figure 4iii). The result for inhalation is a second-order polynomial regression due to the low range and therefore worse resolution in Isensor; a second-order polynomial regression was used to mitigate this problem (Figure 4iv). These equations were programmed into the microcontroller (Figure 1iii) so Spiromni could infer the inhalation and exhalation flow rates.

As the calibration line for exhalation does not intersect at origin, inferred flow rate is more than zero when differential pressure is at zero. It was decided to note the values of flow rate less than 0.5 L/s as 0 L/s (Figure 4iii). Therefore, the resolution of Esensor starts from 0.5 L/s to 10 L/s at steps of 0.01 L/s. Using Spiromni now, critical spirometry values such as Forced Vital Capacity (FVC), Forced Expiratory Volume at 1 s (FEV1), and Peak Expiratory Flow Rate (PEFR) could be measured (Figure 5i,ii). The calibration of Isensor uses a second-order polynomial equation which intersects at the origin and therefore do not face the issue as Esensor (Figure 4iv). It reads between 0 L/s to 2 L/s at a resolution of 0.03 L/s. There is a slight change in resolution as the pressure increases due to the flow rates being inferred from a second-order polynomial fit but it did not significantly affect the reading of Spiromni, as seen in the inhalation profile (Figure 5iii).

## 4. Discussion

The choice to use a medium-size spacer was deliberate to strike a balance between the uniform delivery of a large spacer and portability of a small spacer. Small-volume spacers, although compact, are ineffective in alleviating the problems of coordination that users require to achieve the proper deposition of drugs in their lungs. Large spacers allows for uniform delivery but the chamber size and the number of breaths required to inhale most of the drugs in the large chamber means it is cumbersome to carry around as well as troublesome to use [22].

The Sensirion SDP31 differential pressure sensor used in Spiromni could only perceive differential pressure between 0 Pa to 500 Pa at a resolution of 0.1 Pa, there was a need to adjust the sizes of the PNT screen cutouts in order to measure flow rate ranges required for inhalation and exhalation within SDP31’s limitation. While using umbrella valves over the PNT screen simplified the design, the combination of the valve and the PNT screens resulted in the reading at low flow rates to be inaccurate. This was caused by the umbrella valve unable to fully open at low flow rates and therefore the choice was to eliminate the reading of flow rates below 0.5 L/s in Esensor. While this may cause total expired volume be lowered, it was not perceived to be able to significantly affect the results. The range of flow rate needed to fit within Isensor is smaller but it also had a lower resolution than Esensor. The lower resolution was a result from the PNT screen across Isensor needing an additional consideration: to be large enough to reduce the loss in flow velocity, and thus medication being able to pass through it and into the lungs of the user. The reduction in flow velocity by half from the mouthpiece to the end cap means there is no dead volume within the spacer section where drugs are not being pulled by the user.

FVC, FEV1, FVC/FEV1are parameters in the spirometry component of Pulmonary Function Tests(PFT) (Figure 5i,ii). A low FEV1 is a strong predictor for future asthma attack regardless of symptoms exhibited by patients. The FVC/FEV1ratio a criterion in determining expiratory airflow obstruction by comparing the FEV1 in relation to FVC. PEFR is another indicator of airflow obstruction but it is also important in measuring the effectiveness of the prescribed asthmatic drugs to the user [23]. Inspiration flow (Figure 5iii) can be used to determine if the user is using proper technique and provide feedback. Spiromni currently does not detect the ejection of drugs from the pMDI, from which the optimal window for inhaling the drugs is derived from [24].

The consideration of this paper and the design of Spiromni was a study on the feasibility of measuring both inhalation and exhalation profiles, which was required for both a spirometer and EMD to be placed within a single device. The total cost of the components for Spiromni is calculated to be under USD 120. Other aspects of the device including software, conformation with ISO23747 [25] for respiratory equipment, as well as ISO20072 [26] for aerosol drug delivery devices were not taken into consideration.

Lastly, there is a concern in the public perception that a combination device with multiple functions is perceived to be inferior to a device with single, specialised function even when the attributes and performance are exactly the same [27]. A product built on Spiromni would need to be positioned and marketed strategically.

## 5. Conclusions

In conclusion, the integration of a smart inhaler device that combines a spirometer, EMD, and a spacer represents a significant advancement in respiratory care for asthma patients. Spiromni is a low-cost portable multi-functional device that offers key benefits for both asthma patients and clinicians.

We believe that integrating Spiromni with a dedicated healthcare application can facilitate timely communication, allowing healthcare providers to monitor patients remotely, track adherence, and manage care more effectively to tailor personalised treatment plans.

## Figures and Tables

**Figure 1 bioengineering-11-00552-f001:**
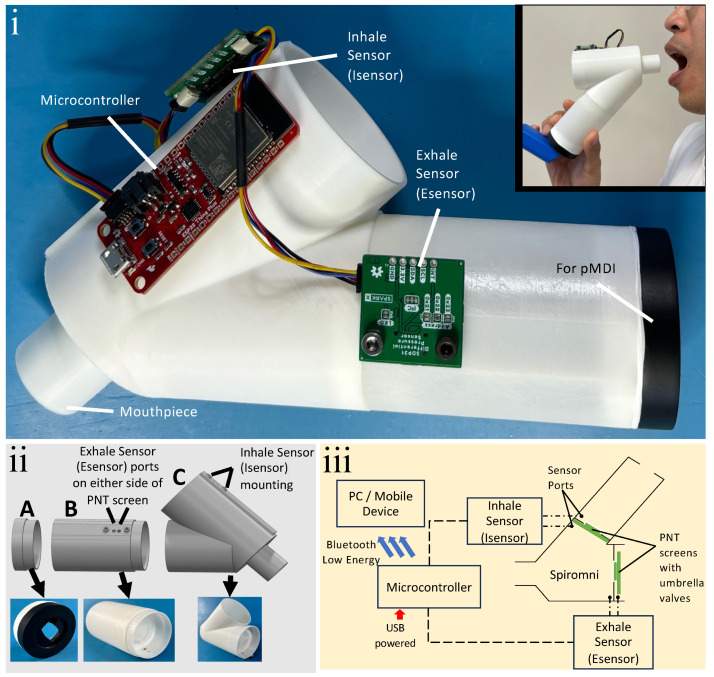
(**i**) Photo of spacer device with integrated spirometer and EMD, Spiromni; (inset) photo of a user with Spiromni. (**ii**) Exploded 3D model of Spiromni and its 3 sections: (A) end cap with silicone sheet cut to shape of the pMDI mouthpiece adhered with silicone sealant; (B) spacer section with 100 mL capacity; (C) spirometer section with mouthpiece. (**iii**) Schematic of Spiromni: each Sensiron SDP31 differential pressure sensor (Isensor and Esensor) has 2 sensor ports and they measure across the PNT screens with integrated umbrella valves.

**Figure 2 bioengineering-11-00552-f002:**
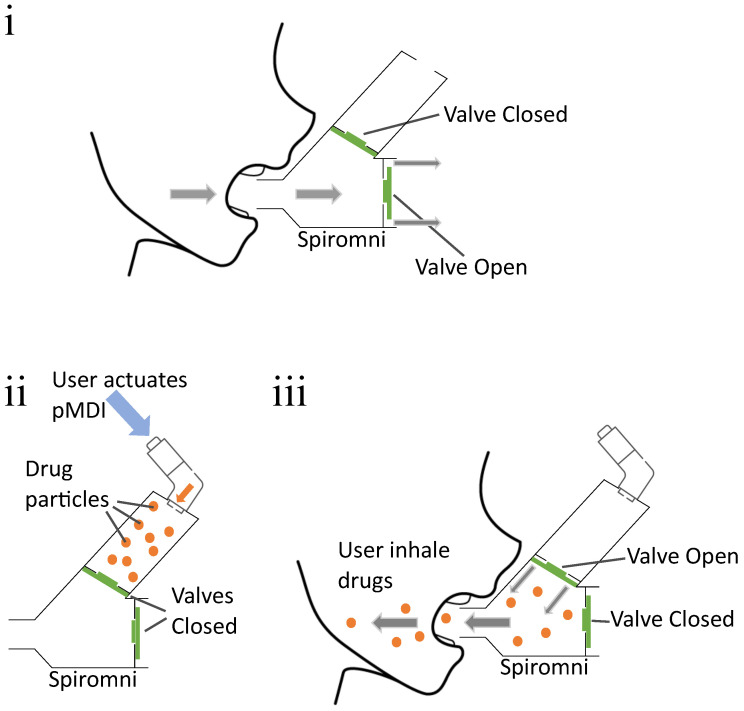
(**i**) Flow diagram for exhalation: User takes a deep breath and exhales forcefully through Spiromni. The umbrella valves diverts the air flow through the bottom part of the spirometer section. (**ii**,**iii**) Flow diagram for inhalation: User actuates pMDI to eject drugs into spacer section, then actuates the umbrella valve with their inhalation to breathe in drugs from the spacer section.

**Figure 3 bioengineering-11-00552-f003:**
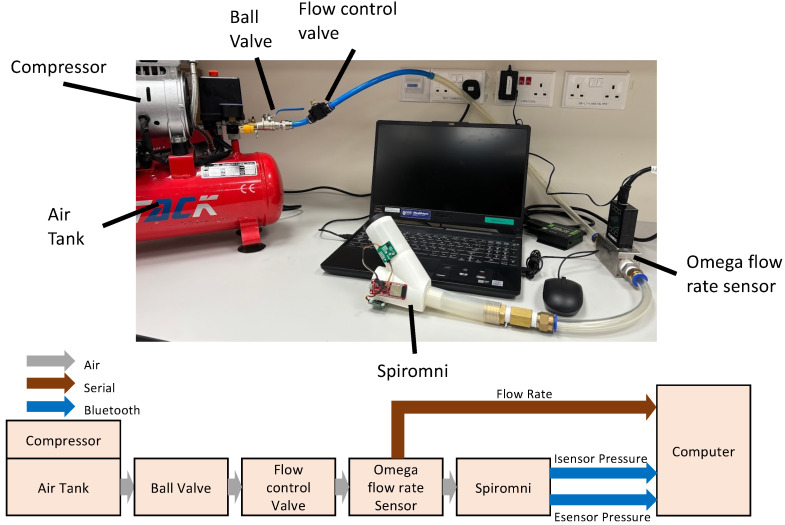
Sensor calibration setup (top) and schematic (bottom).

**Figure 4 bioengineering-11-00552-f004:**
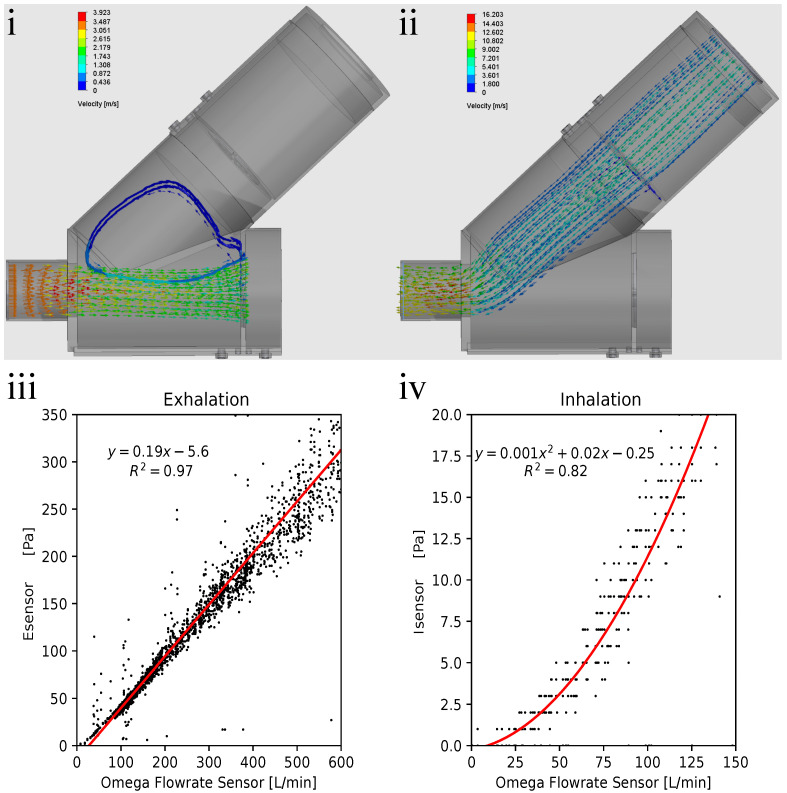
(**i**) Exhalation flow simulation results: turbulence in the spirometer section could be observed at the intersected cylinders (blue arrows) resulting in an 8% loss in flow velocity. (**ii**) Inhalation flow simulation results. Flow velocity is 88% at Isensor and 57% at the end cap from the mouthpiece, allowing the user to pull drugs across the entirety of the spacer section in a laminar way. (**iii**) Exhalation flow rate–pressure graph, showing fit line in red, the equation, and $r^{2}$ value. Line does not intersect at origin due to the umbrella valves over the PNT screen not fully opening at low flow rates. (**iv**) Inhalation flow rate–pressure graph, with second-order polynomial fit line in red, plus the equation and $r^{2}$ value on the top left of the graph.Polynomial fit is used to mitigate issues with resolution in the lower range of flow rate measured for inhalation.

**Figure 5 bioengineering-11-00552-f005:**
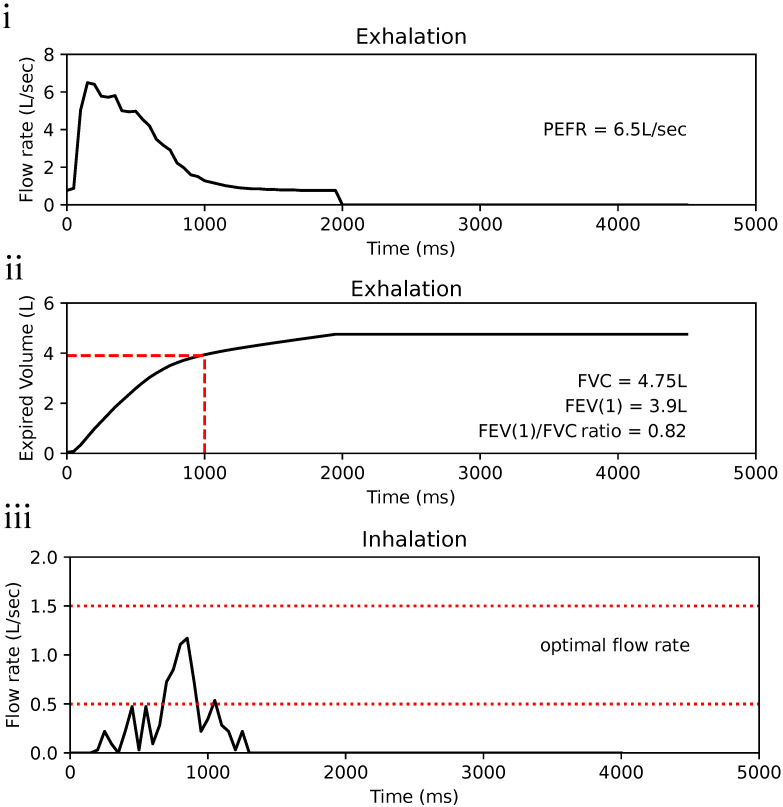
(**i**) Exhalation profile in flow rate–time to show peak expiratory flow rate (PEFR), value of PEFR on the right of graph. (**ii**) Expired volume graph showing Forced vital capacity (FVC) and Forced expiratory volume in the first second (FEV1), enabling the Tiffeneau–Pinelli index (FVC/FEV1) to be calculated. Values are again on the right of the graph. (**iii**) Inhalation profile of a user using Spiromni with a pMDI: the section in between the red dotted lines is the recommended flow rate for the optimal deposition of medication into the lungs [17]. Spiromni currently does not detect the ejection of medication from pMDI.

## Data Availability

Dataset available on request from the authors.

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
