# Peer review of "Metered-Dose Inhaler Spacer with Integrated Spirometer for Home-Based Asthma Monitoring and Drug Uptake"

_bioengineering, 2024, doi:10.3390/bioengineering11060552_

Round 1

Reviewer 1 Report

Comments and Suggestions for Authors

This paper describes a novel asthma management medical device. The Spiromni is friendly to commercialization and would be very helpful in clinics. To make this paper better and more effectively demonstrate the benefits of the suggested device, I have a few suggestions. 

1. The authors are able to offer a table that compares the proposed device with products available in stores.  The paper claims that the low cost of the suggested device is one of its advantages, but it is difficult to find evidence to support this claim. The comparison table may help reduce public bias regarding the multifunctional combination device that was mentioned in the discussion section.

2. The authors noted in the results section that "to minimize turbulence as much as possible, the flow pattern within Spiromni during exhalation and inhalation were constantly monitored during the design process." Could you describe the strategy you followed to reduce turbulence? The strategy could be an important contribution in this paper.

Reviewer 2 Report

Comments and Suggestions for Authors

This submission paper describes the development and the design of a single, multi-functional device named Spiromni, which combines three different pressurized metered dose inhaler (pMDI) accessories such as a pMDI spacer, an electronic monitoring device (EMD), and a spirometer. The device is designed for the home-based asthma monitoring and drug uptake.

The following comments could be considered to improve the manuscript:

1) Some details of the accessories should be provided, for example, the model information of the EMD, Esensor, the microchip in the microcontroller.

2) The details of the flow profile simulation should be provided. Which mathematical model was used for simulation? Which parameters were used in the simulation, or how to determine these parameters? 

3) Regarding the curve fitting results, despite of the R values, it is necessary to interpret the meaning of model parameters. For example, the linear regression equation of exhalation in Figure 4(iii) is y=0.19x-5.6, so if the value of Omega flowrate sensor is zero, then the Esensor value would be -5.6 instead of 0. Is that reasonable? Please explain and interpret such results. The same phenomenon occurs in inhalation. Why using a quadratic polynomial function for regression? Please provide the reason for such a consideration.

4) It would be necessary to clarify which international standards or industry standards (if possible) the designed product comply with? And which parameters or testing results of the product meets the standards.

Round 2

Reviewer 2 Report

Comments and Suggestions for Authors

The revised manuscript has been improved in response to the review comments.

Author Response

  1. The authors made a comparative table requested by reviewer 1, which they have not included in the manuscript (it is in the response letter to the reviewer). However, I think it would be important to include it in the manuscript as it would provide interesting information for readers.

We have referenced examples of some of the products such as the Teva Digihaler, MIR Spirobank in the introduction. However, we prefer not to include their costs in the main article we found the prices vary across regions. The official websites of these products are include in the references and readers can refer to them for more information.

  1. Also, it would be interesting to include the main considerations when designing Spiromni. Perhaps as Supporting information.

We have included the main considerations in section 2.1 Spiromni.

“The two main considerations when designing Spiromni were: (a) accommodate the range of the inhalation and exhalation flowrates for sensors measurement, and (b) to minimize the turbulence and dead volume for accurate flow rate measurements, as well as to optimise drug delivery from the spacer to the user.”
